# Enhancing Muscle Intracellular Ca^2+^ Homeostasis and Glucose Uptake: Passive Pulsatile Shear Stress Treatment in Type 2 Diabetes

**DOI:** 10.3390/biomedicines11102596

**Published:** 2023-09-22

**Authors:** Arkady Uryash, Jordan Umlas, Alfredo Mijares, Jose A. Adams, Jose R. Lopez

**Affiliations:** 1Division of Neonatology, Mount Sinai Medical Center, Miami, FL 33140, USA; auryash@msmc.com (A.U.); jose.adams@msmc.com (J.A.A.); 2Centro de Biofísica y Bioquímica, Instituto Venezolano de Investigaciones Científicas, Caracas 21827, Venezuela; mijaresa@gmail.com; 3Department of Research, Mount Sinai Medical Center, Miami Beach, FL 33140, USA

**Keywords:** type 2 diabetes, pGz, intracellular Ca^2+^, glucose uptake, intracellular ROS, eNOS, nNOS, iNOS, muscle function

## Abstract

Type 2 diabetes mellitus (T2D) is a significant global public health problem that has seen a substantial increase in the number of affected individuals in recent decades. In a murine model of T2D (db/db), we found several abnormalities, including aberrant intracellular calcium concentration ([Ca^2+^]_i_), decreased glucose transport, increased production of reactive oxygen species (ROS), elevated levels of pro-inflammatory interleukins and creatine phosphokinase (CK), and muscle weakness. Previously, we demonstrated that passive pulsatile shear stress, generated by sinusoidal (headward–forward) motion, using a motion platform that provides periodic acceleration of the whole body in the Z plane (pGz), induces the synthesis of nitric oxide (NO) mediated by constitutive nitric oxide synthase (eNOS and nNOS). We investigated the effect of pGz on db/db a rodent model of T2D. The treatment of db/db mice with pGz resulted in several beneficial effects. It reduced [Ca^2+^]_i_ overload; enhanced muscle glucose transport; and decreased ROS levels, interleukins, and CK. Furthermore, pGz treatment increased the expression of endothelial nitric oxide synthase (eNOS), phosphorylated eNOS (p-eNOS), and neuronal nitric oxide synthase (nNOS); reduced inducible nitric oxide synthase (iNOS); and improved muscle strength. The cytoprotective effects of pGz appear to be mediated by NO, since pretreatment with L-NAME, a nonspecific NOS inhibitor, abolished the effects of pGz on [Ca^2+^]_i_ and ROS production. Our findings suggest that a non-pharmacological strategy such as pGz has therapeutic potential as an adjunct treatment to T2D.

## 1. Introduction

Type 2 diabetes (T2D) is a metabolic disorder characterized by elevated blood glucose levels and muscle insulin resistance, which can cause various complications, such as renal failure, neuropathy, and cardiac dysfunction [1]. Skeletal muscle is a critical site for glucose uptake, and impairments in its function (insulin resistance) constitute hallmarks of T2D [2]. The mechanisms that underlie this condition are not yet fully understood [3], but a potential contributor could be alterations in muscle [Ca^2+^]_i_ [4].

Calcium plays a multifaceted role in the regulation of muscle function [5]. The concentration of Ca^2+^ within cells is tightly controlled through a combination of mechanisms involving the removal of Ca^2+^ from the cell and sequestration by intracellular organelles [6]. This helps to maintain a low concentration of [Ca^2+^]_i_ of around 100–120 nM, while the Ca^2+^ outside the cell is on the order of mM [4]. Recently, our research group demonstrated in two experimental models that muscles with abnormal [Ca^2+^]_i_ exhibit a reduction in glucose-dependent insulin transport and that reducing [Ca^2+^]_i_ transforms muscles from insulin-resistant to non-insulin-resistant, with a subsequent improvement in hyperglycemia [4].

Despite a new generation of medications and advances in clinical treatments, glycemic control and cardiovascular complications still represent problems for diabetic patients. Whole-body periodic acceleration (pGz) is the sinusoidal motion of the supine body headwards and forwards using a motion platform. This motion passively imparts low-amplitude pulses to the vascular endothelium, thus increasing pulsatile shear stress on the endothelium and causing increase nitric oxide (NO) bioactivity [7]. NO is synthesized in a variety of tissues from l-arginine by a group of enzymes known as nitric oxide synthases (NOS) [8]. There are three isoforms of NOS: neuronal NOS (nNOS); endothelial NOS (eNOS), referred to as constitutive; and inducible NOS (iNOS). Depending on the isoform involved in NO synthesis, it can exert either anti-inflammatory effects (at nanomolar concentrations, mediated by eNOS and nNOS) or pro-inflammatory effects (at micromolar concentrations, mediated by iNOS). Pulsatile shear stress, as produced by pGz, has been shown to upregulate the protein expression of both nNOs and eNOS, thus increasing NO synthesis and increasing antioxidant expression, including superoxide dismutase, catalase, and total antioxidant capacity [7]. Furthermore, pGz has been shown to significantly decrease [Ca^2+^]_i_ in muscle cells with abnormal intracellular [Ca^2+^] [9].

We hypothesize that the passive pulsatile shear stress produced by pGz increases the expression of eNOS and nNOS; reduces elevated [Ca^2+^]_i_; enhances muscle glucose uptake; alleviates abnormal intracellular ROS generation and calpain activity; and decreases muscle injury, pro-inflammatory cytokines, and iNOS, which in turn may lead to an improvement in strength in a T2D mouse model.

## 2. Materials and Methods

### 2.1. Animal Model

In this study, we used male and female wild-type (WT) normoglycemic mice (C57BL6/J) and diabetic db/db mice (Jackson Laboratory, Bar Harbor, ME, USA) at 6 months of age. The experimental animals were provided with food and water ad libitum and kept on a 12 h light/dark cycle. After a week of acclimatization, the WT and db/db mice were randomly divided into four groups: (i) WT control, which did not receive any treatment; (ii) WT-pGz, which received treatment for 8 days (see pGz protocol); (iii) db/db control, which did not receive treatment; and (iv) db/db pGz, treated with pGz (see Section 2.2). The protocol was approved by the Mount Sinai Medical Center Institutional Animal Research Committee under approval codes P-20-05-A-03 and P-23-09-A03, with approval dates of 03/12/2020 and 03/27/2023, respectively.

### 2.2. pGz Protocol

Unanesthetized mice were placed into a rodent holder (Kent Scientific, Torrington, CT, USA), which was placed on a motion platform (SK-L180-Pro, Scilogex, CT, USA). The platform was set to a frequency of 480 cycles per minute and Gz ± 3.0 m/s^2^ for 1 h per day (morning 9–11 a.m.) for 8 consecutive days, as previously described (Adams 2009 [7]). The WT and db/db control groups were also placed in the rodent holder for 1 h per day for 8 consecutive days, but did not receive pGz treatment.

### 2.3. Double-Barreled Ca^2+^-Selective Microelectrodes

Double-barreled Ca^2+^-selective microelectrodes were prepared from thin-walled borosilicate with 1.2 and 1.5 mm outside diameter (OD) (PB150F-4, World Precision Instruments, Sarasota, FL, USA). The ion-selective barrel (1.5-mm OD) was silanized with dimethyldichlorosilane vapor and then backfilled with the Ca^2+^ ionophore II (ETH 129) Sigma-Aldrich in Saint Louis, MO, USA. The remaining portion of the ion-selective barrel was backfilled with pCa7, as previously described in [10]. The resting membrane potential (RMP) barrel (1.2-mm OD) was backfilled with 3 M KCl before the measurements. The RPM- and Ca^2+^-specific potentials were acquired at a frequency of 1000 Hz using AxoGraph software (version 4.6; Axon Instruments, San Jose, CA, USA), then and stored on a computer for further analysis. Before and after each measurement, individual calibration of the Ca^2+^-selective microelectrodes was performed by following the previously described protocol [10]. If the calibration curves obtained before and after the measurement differed by more than 3 mV, the data from that microelectrode were excluded from further analysis [10].

### 2.4. Recording of [Ca^2+^]_i_ and Glucose Uptake in Muscle Fibers In Vivo

pGz-treated and untreated wild-type (WT) and diabetic (db/db) mice were anesthetized with 100 mg/kg of ketamine and 5 mg/kg of xylazine. They were then placed into a heating system (ATC1000, World Precision Instruments, FL, USA) to maintain a constant body temperature of 37 °C [10]. Hair was removed from the leg (left or right), and an incision was made over the gastrocnemius muscle. The superficial muscle fibers were exposed and kept moist by perfusing them with a warm Ringer solution. To assess muscle [Ca^2+^]_i_ and glucose uptake in vivo, muscle cells were incubated with a fluorescent glucose analog called 2-(N-(7-Nitrobenz-2oxa-1,3-diazol-4yl)Amino)-2-Deoxyglucose (2-NBDG) at a concentration of 600 µM in a glucose-free Ringer solution [4]. Thirty minutes after mice received IP insulin injections of 0.75 IU/kg body weight (Novo Nordisk Inc., Plainsboro, NJ, USA), muscle cells loaded with 2-NBDG were impaled with the Ca^2+^-selective microelectrode, and the rate of increase in the fluorescence of 2-NBDG was recorded and stored on a computer for future analysis [4]. The 2-NBDG was excited at 920 nm and collected at a wavelength of 540 nm. The experiments were carried out at 37 °C.

### 2.5. Determination of ROS in Muscle Cells

Intracellular levels of ROS were evaluated using the dichlorodihydrofluorescein diacetate (DCFHDA) assay (Saint Luis, MO, USA) in vivo on superficial gastrocnemius muscle cells from WT and db/db treated and untreated mice. An excitation wavelength of 488 nm and an emission wavelength of 525 nm were used. The results were normalized to untreated WT muscle cell values.

### 2.6. Creatine Kinase Serum Level Determination

Blood samples were obtained from WT and db/db mice by tail cut. Serum creatine kinase (CK) activity was assessed in pGz-treated and -untreated mice using the UV-kinetic method (Teco Diagnostics, Anaheim, CA, USA), following the manufacturer’s instructions. CK enzymatic activity is expressed as Kilo International Units per liter (KIU/L).

### 2.7. Measurement of Plasma Interleukin and TNF-α Concentrations

Plasma IL-6 concentrations were assessed in plasma samples collected from untreated and pGz-treated animals using the Milliplex Mouse Cytokine/Chemokine Panel (EMD Millipore, Burlington, MA, USA) in conjunction with the Bio-Plex Suspension Array System (Bio-Rad Laboratories, Hercules, CA, USA). Similarly, TNF-α levels were measured in plasma samples from the same animals using a mouse TNF-α ELISA kit (Invitrogen, Carlsbad, CA, USA). The values were normalized to the muscle protein concentration determined by the BCA protein assay (Thermo Scientific, Waltham, MA, USA). All measurements were made following the instructions provided by the respective manufacturers.

### 2.8. Muscle Functional Testings

Two different functional tests were performed in WT and db/db mice: (i) the two-limb hanging wire test and (ii) the forelimb grip strength. Two-limb hanging wire test: Mice were randomly assigned to four groups: (i) untreated WT control mice (*n*_mice_ = 8); (ii) WT control mice subjected to the pGz protocol (*n*_mice_ = 10); (iii) untreated db/db mice (*n*_mice_ = 10); and (iv) db/db mice treated with pGz (*n*_mice_ = 13). pGz-treated and -untreated mice were gently placed onto a metallic wire (2 mm thick, 55 cm long) tightly secured between 2 vertical poles, allowing them to grasp the wire with their forelimbs to suspend their body weight. This test was based on the latency of a mouse falling off a metal wire upon exhaustion. As soon as the mouse was suspended properly, the timer started. The test was performed three times for each mouse (5 min intervals) before and after pGz treatment, and the mean hanging time was recorded and stored for future analysis. Forelimb grip strength. Forelimb grip strength was measured before and after pGz treatment in WT and db/db mice using a stainless steel clamp to assess muscle strength (Columbus Instruments, Columbus, OH, USA). Mice were randomly assigned to one of four groups: (i) untreated WT control mice (*n_mice_* = 10); (ii) WT control mice subjected to the pGz protocol (*n_mice_* = 10); (iii) untreated db/db mice (*n_mice_* = 11); and (iv) db/db mice treated with pGz (*n_mice_* = 11). Briefly, the pGz-treated and -untreated mice were grasped by the tail and gently placed (i) onto the wire (two-limb hanging wire test), allowing their forepaws to grasp the wire, and the hanging time was measured. (ii) The grip (forelimb grip strength) was assessed, and then the mice were smoothly pulled backward by the tail until their grip was released. Each animal performed three independent trials (5 min recovery period) before and after pGz treatment. The three measurements were averaged and, in the case of forelimb grip strength, were divided by the body weight in grams. To minimize variability, both tests were performed under identical conditions, including room type, temperature, occupancy, and time of day (8–10 a.m.).

### 2.9. N-nitro-L-arginine Methyl Ester (L-NAME) Protocol

L-NAME is a non-selective inhibitor of NO-synthase used to induce NO synthesis deficiency [11]. A different db/db mouse cohort was randomized into three groups to investigate the effects of L-NAME and pGz-induced cellular protection. The three groups in the study were as follows: (A) db/db receiving no treatment (*n_mice_* = 3); (B) db/db pGz-treated for 8 consecutive days (*m_mice_* = 4); and (C) db/db mice pre-treated with L-NAME for 7 days and then treated with pGz for 8 consecutive days (*m_mice_* = 6). L-NAME was administered to the mice by providing it in their drinking water at a concentration of 1 g/L, equivalent to a dose of 180 mg/kg. After completing the treatment protocol, [Ca^2+^]_i_ and ROS production were measured in vivo for all groups. These markers were used as indicators of cellular protection induced by pGz.

### 2.10. Glucose Determinations

Blood samples for glucose analysis were obtained from fasting mice through a small cut at the end of the tail; glucose measurements were made using a glucometer (AlphaTRAK^®^ glucose monitor, Abbott Animal Health, Abbott Park, IL, USA). Mice with fasting blood glucose levels > 250 mg/dL were considered diabetic.

### 2.11. Western Blotting and Protein Expression

Another cohort of WT and db/db mice treated with and without pGz was anesthetized (100 mg/kg of ketamine and 5 mg/kg of xylazine) and sacrificed by cervical dislocation. Gastrocnemius muscles were dissected, minced, and homogenized using a total protein extraction kit (Sigma Millipore, Saint Louis, MA, USA). Total protein concentrations were determined using the bicinchoninic acid (BCA) method (Thermo-Scientific, Waltham, MA, USA). Denatured, SDS-gel separated, and membrane-immobilized proteins were incubated overnight at 4 °C with primary antibodies: anti-eNOS, dilution 1:2500 (ab300072; Abcam, Waltham, MA, USA); anti-p-eNOS, dilution 1:2500 (ab230158, Abcam, MA, USA); anti-nNOS, dilution 1:2000 (ab76067, Abcam, MA, USA); anti-iNOS, dilution 1:2500 (ab283655, Abcam, MA, USA); anti-GAPDH, dilution of 1:5000 (SC47724; Santa Cruz, CA, USA); and secondary fluorescent antibodies (Abcam, MA, USA). The resolved bands were detected with a Storm 860 Imaging System (GE Bio-Sciences, Piscataway, NJ, USA). Protein levels were quantified using myImageAnalysis software V1.0 (Thermo-Fisher Scientific, Waltham, MA, USA) and normalized to glyceraldehyde-3-phosphate dehydrogenase (GAPDH).

### 2.12. Solutions

The mammalian Ringer solution used in this experiment contained the following concentrations (in mM): 130 NaCl, 5 KCl, 1.8 CaCl_2_, 1 MgCl_2_, 5 glucose, and 10 HEPES, with a pH of 7.4. The composition of the glucose-free Ringer solution was identical to that of the mammalian Ringer solution, but glucose was removed and mannitol was added to maintain osmolarity. The solutions were continuously bubbled with 95% O_2_ and 5% CO_2_.

### 2.13. Statistical Analysis

Data are presented as the mean ± standard deviation (SD). Muscle fibers with a resting membrane potential of less than −80 mV were excluded. To determine the normal distribution of the samples, the D’Agostino and Pearson test was utilized. For multiple measurements, one-way ANOVA followed by Tukey’s post-test was performed. The number of muscle cells in which successful measurements were performed is represented by *n*, while *n_mice_* represents the total number of mice used in the study. A *p*-value of less than 0.05 was considered significant. GraphPad Prism 10 software (GraphPad Software, Inc., San, Diego, CA, USA) was used for statistical analysis.

## 3. Results

### 3.1. pGz Reduced [Ca^2+^]_i_ and Increase Glucose Uptake in db/db Skeletal Muscle

Chronic elevation of [Ca^2+^]_i_ in the skeletal muscle has been found to negatively impact glucose tolerance [4]. To assess the effect of pGz on [Ca^2+^]_i_ and glucose uptake, we simultaneously measured [Ca^2+^]_i_ and glucose uptake in vivo in superficial gastrocnemius fibers of anesthetized WT and db/db. [Ca^2+^]_i_ was measured using double-barreled Ca^2+^-selective microelectrodes [4], and glucose uptake was measured with the fluorescent glucose sensor 2-NBDG [12,13]. We confirmed a significant increase in resting [Ca^2+^]_i_ and a marked reduction in insulin-induced glucose uptake in db/db muscle compared to WT [4]. [Ca^2+^]_i_ was 160% more elevated in db/db than in WT (318 ± 51 nM versus 122 ± 3 nM, *p* < 0.001) (Figure 1A), and glucose uptake was reduced 64% in db/db compared to WT muscles (*p* < 0.001 compared to WT (Figure 1B). Notably, pretreatment of db/db mice with pGz (1 h per day, for 8 consecutive days) provoked a reduction in [Ca^2+^]_i_ of 50% (159 ± 20 nM; *p* < 0.001 compared to untreated db/db mice) and an elevation of 121% in glucose uptake (*p* < 0.001 compared to untreated db/db mice) (Figure 1A,B). No significant effect on [Ca^2+^]_i_ and glucose uptake was observed in the WT group.

### 3.2. Effect of pGz on ROS Production in db/db Muscle Fibers

Accumulating evidence has been presented that high glucose in diabetes increases ROS production, which can lead to tissue injury [14]. Therefore, ROS production was measured in muscle cells from WT and db/db mice. The db/db muscle fibers showed a significant increase in ROS levels of 157% compared to WT muscle (*p* < 0.001) (Figure 2). Pretreatment with pGz significantly reduced ROS production (39%) in db/db muscle cells (*p* < 0.001 compared to untreated db/db mice) (Figure 2). No effect of pGz was detected in WT muscle cells (*p* = 0.88 compared to untreated WT mice).

### 3.3. Reduction in Muscle Injury by pGz in db/db Mice

To investigate the hypothesis that pGz can reduce muscle damage in db/db mice, we measured the CK levels before and after pGz treatment. A clear difference in the severity of the muscle damage was found between WT and db/db muscles. Figure 3 shows that db/db mice exhibited significantly elevated serum CK levels compared to WT mice (6 ± 0.9 KIU/L vs. 0.61 ± 0.1 KIU/L; *p* < 0.001 compared to WT mice). However, pGz treatment resulted in a significant reduction in CK levels in db/db mice (6.29 ± 1.5 KUI/L; *p* < 0.001 compared to untreated db/db mice) (Figure 3). No significant pGz effect was observed in WT mice (*p* = 0.99).

### 3.4. Effects of pGz on Plasma Interleukin 6 and TNF-α Concentrations in db/db Mice

Proinflammatory IL-6 is a critical contributor to insulin resistance, and consequently has been implicated in the pathophysiology of T2D [15]. Therefore, IL-6 was measured in plasma from WT and db/db mice. We found that the concentration of IL-6 increased from 19 ± 3 pg/mL in WT to 46 ± 9 pg/mL in db/db mice (*p* < 0.001 compared to WT mice) (Figure 4A). Treatment with pGz significantly reduced the plasma levels of IL-6 in db/db mice to 27 ± 6 pg/mL (*p* < 0.001 compared to untreated WT db/db mice) (Figure 4A). In the WT group, pGz did not reduce the plasma IL-6 level (20 ± 3 pg/mL (*p* = 0.98 compared to untreated WT mice) (Figure 4A). TNF-α plays a pivotal role in the regulation of insulin resistance and endothelial dysfunction [15,16]. Consequently, the TNF-α level was measured in the plasma of WT and db/db mice. Similarly, the TNF-α level was significantly increased in db/db mice to 14.2 ± 2.1 pg/mg protein versus 5.3 ± 1.2 pg/mg protein in WT mice (*p* < 0.001 compared to WT mice) (Figure 4B). Likewise, pGz treatment significantly reduced the TNF-α levels in db/db mice to 8.8 ± 1.1 pg/mg protein (*p* < 0.001 compared to untreated db/db mice) (Figure 4B). No significant effect was observed in young mice (4.9 ± 0.7 pg/mg^−1^ protein (*p* = 0.88 compared to untreated WT mice).

### 3.5. pGz Improve Muscle Funtion in Diabetic Mice

Fatigue-related symptoms, such as reduced endurance, are commonly reported by individuals with T2D [17,18]. Therefore, we studied the effect of pGz using two functional tests (two-limb hanging wire test and forelimb grip strength) in control and db/db mice before and after treatment. db/db mice exhibited reduced muscle function compared to non-diabetic mice. Specifically, the hanging time of db/db mice was reduced by 39% (31 ± 4 s, versus 51 ± 6 s) compared to WT mice (*p* < 0.001) (Figure 5A). Treatment with pGz improved the performance by 32% in db/db mice (41 ± 5 s, *p* < 0.01 compared to untreated mice); however, no significant effect was observed in the WT group (*p* = 0.94 compared to untreated WT mice) (Figure 5A). In db/db mice, the forelimb grip strength was reduced by 39% (*p* < 0.01 compared to WT) (Figure 5B). However, pGz treatment improved the muscle strength of db/db mice by 31% (Figure 5B). In contrast, no significant effect was observed in WT mice after treatment with pGz (*p* = 0.94 compared to untreated WT mice).

### 3.6. The Role of NO in the Cytoprotective Effects of pGz

The cytoprotective effects of pGz on [Ca^2+^]_i_ and intracellular ROS production in db/db muscle cells were abolished when db/db mice were pretreated with L-NAME, a nonselective inhibitor of NOS (Figure 6A). The inhibition of NOS in db/db mice resulted in a significant increase in [Ca^2+^]_i_ of 64% (*p* < 0.001 compared to untreated L-NAME db/db mice) and an increase in ROS generation of 174% (*p* < 0.001 compared to untreated L-NAME db/db mice) (Figure 6B).

### 3.7. Effect of pGz on Blood Glucose Level

The average basal glucose value in fasting WT mice was 97 ± 4 mg/dL (*n* = 16 and *n_mice_* = 8), while in db/db mice, it was 350 ± 30 mg/dL (*p* < 0.001 compared to WT mice) (Figure 7). After receiving pGz treatment, the glucose values in db/db mice decreased significantly to 192 ± 36 mg/dL (*p* < 0.001 compared to untreated db/db mice), while pGz treatment did not affect the glucose values in WT mice (*p* = 0.99).

### 3.8. Effect of pGz on the Expression of the NOS Protein in db/db Muscle

Protein expression analysis of the db/db muscle revealed noteworthy changes in NOS expression. db/db muscles showed reductions of 50% and 48% in the expression of eNOS and p-eNOS, respectively (*p* < 0.001 compared to WT) (Figure 8A,B). However, no significant differences were observed in nNOS expression between the db/db and WT muscle (*p* = 0.95). Interestingly, iNOS expression in db/db muscle was significantly elevated by 671% compared to WT (*p* < 0.001) (Figure 8A,B). Upon the administration of pGz treatment, there was a clear effect on the expression of NOS and p-eNOS in both genotypes. In pGz-treated db/db skeletal muscle, eNOS increased by 66% (*p* < 0.01 compared to untreated db/db) and p-eNOS by 64% (*p* < 0.01 compared to untreated db/db). In WT mice, pGz treatment increased eNOS expression by 33% (*p* < 0.001 compared to untreated WT) and p-eNOS by 58% (*p* < 0.001 compared to untreated WT). pGz treatment increased nNOS expression by 17% in db/db (*p* < 0.05 compared to untreated db/db) and 61% in WT muscles (*p* < 0.01 compared to untreated WT). Remarkably, pGz caused a 49% reduction in iNOS expression in db/db muscle (*p* < 0.01 compared to untreated db/db) and 8% in WT (*p* < 0.05 compared to untreated WT).

## 4. Discussion

The present study aimed to investigate the effects of passive pulsatile shear stress (pGz) on skeletal muscle in db/db mice, a genetic model of T2D with insulin resistance.

The significant findings were:The application of pGz effectively reduced the [Ca^2+^]_i_ overload and increased the glucose uptake observed in db/db skeletal muscle fibers;The study revealed that db/db muscle exhibited elevated intracellular ROS production, and treatment with pGz effectively reduced this elevated ROS production;The serum levels of CK in db/db mice were significantly higher compared to WT mice, and pGz intervention reduced CK levels in db/db mice;There was a significant increase in pro-inflammatory cytokines, specifically IL-6 TNF-α, in db/db mice; pGz treatment decreased the plasma concentrations of both IL-6 and TNF-α in db/db mice;db/db muscles showed greater fatigability and reduced force compared to WT; pGz treatment resulted in a reduction of muscle fatigability and improvement of muscle force;L-NAME pretreatment abolished the cytoprotective effect of pGz on [Ca^2+^]_I_ and ROS production in db/db mice;The expression of eNOS and p-eNOS proteins in db/db muscle was reduced, while iNOS was enhanced. Treatment with pGz reversed the expression pattern observed in db/db mice by increasing eNOS, p-eNOS, and nNOS expression while causing a decrease in iNOS levels in db/db skeletal muscle.

T2D is a multifaceted and multifactorial condition that comprises a significant portion of the diabetic population [19,20,21]. The prevalence of T2D has increased throughout all regions of the world during the last three decades [22].The complex interplay of genetic and environmental factors contributes to the development of T2D, making it one of the leading causes of death among adults on a global scale [22]. In healthy humans, 70–80% of glucose uptake occurs in the skeletal muscle in the postprandial state through both insulin-dependent and insulin-independent glucose pathways, where skeletal muscle plays a vital role in maintaining glucose homeostasis [23]. One of the key underlying factors of T2D is the reduced sensitivity of the skeletal muscle cells to insulin, known as insulin resistance, which is characterized by decreased responsiveness of the body’s cells to insulin, resulting in impaired glucose uptake and elevated blood sugar levels [24,25,26,27]. We have shown that muscle insulin resistance is related to a chronic increase in [Ca^2+^]_i_, which has an inhibitory effect on glucose uptake in humans and rodents [4].

### 4.1. pGz Reduces Intracellular Ca^2+^ and Improves Glucose Uptake

In the current study, using a db/db (leptin-receptor-deficient) mouse model, a well-stablished model for studying T2D, we confirmed our previous finding that db/db skeletal muscle exhibited chronic elevation of [Ca^2+^]_i_ and reduced glucose uptake compared to non-diabetic muscles [4]. The intracellular concentration of Ca^2+^ in skeletal muscle is accurately controlled through intricate spatial and temporal regulatory mechanisms that imply a delicate balance between the influx and release of Ca^2+^ from intracellular organelles, as well as sequestration within the cell and extrusion to the extracellular environment [6,28,29]. In quiescent muscle cells, [Ca^2+^]_i_ is maintained at a relatively low level, typically ranging from 100 to 120 nanomolars [4]. pGz treatment significantly reduced [Ca^2+^]_i_ and improved insulin-induced glucose uptake in the skeletal muscle of db/db mice, and subsequently led to a decrease in the blood glucose levels of fasting db/db mice. The mechanism by which pGz causes the reduction in [Ca^2+^]_i_ is not fully understood; however, it is possible that NO production induced by pGz treatment (i) decreases Ca^2+^ release from the sarcoplasmic reticulum by a reduction in the open probability of the ryanodine receptor [30,31]; (ii) modifies the degree of nitrosylation of the ryanodine receptor, reducing the leak of Ca^2+^ from intracellular stores (unpublished observations); or (iii) reduces the influx of Ca^2+^ through the muscle sarcolemma [32,33]. The effect of pGz on [Ca^2+^]_i_ was mediated by nitric oxide since the pretreatment stage for 8 days, with L-NAME, a NOS inhibitor [34,35], abolishing the effect elicited by pGz on [Ca^2+^]_i_.

Abnormal insulin-stimulated glucose transport has been observed in fat cells [36,37] and muscle [4] with elevated [Ca^2+^]_i_. Lowering [Ca^2+^]_i_ levels in muscle cells with persistently high [Ca^2+^]_i_ transforms them from insulin-resistant to non-insulin-resistant, leading to subsequent improvements in hyperglycemia [4]. However, the precise mechanism(s) by which elevated [Ca^2+^]_i_ contributes to insulin resistance are not completely understood. pGz enhances glucose uptake in db/db muscles, which could be attributed to its ability to reduce [Ca^2+^]_i_ and/or stimulate non-insulin NO-mediated glucose transport [38]. 

### 4.2. Intracellular ROS Is Reduced by pGz in db/db Mice

ROS production was significantly elevated in the db/db muscle, in concordance with a previous report [39]. T2D is frequently associated with increased ROS, resulting in oxidative stress [40]. This increase in ROS production reduces NO’s bioavailability [41] and can trigger apoptotic pathways [42]. Excess ROS production can overwhelm the antioxidant system’s capacity to neutralize oxidative stress, leading to skeletal and cardiac dysfunctions [43]. However, when the db/db mice were subjected to pGz treatment, a significant reduction in ROS production was observed. This effect could be mediated by an increase in endogenous antioxidant expression, such as glutathione peroxidase-1, catalase, superoxide, and superoxide dismutase 1, induced by pGz [7]. Our study’s findings highlight ROS’s involvement in T2D and the potential therapeutic benefits of pGz treatment. By reducing ROS production in db/db striated muscle, pGz treatment may help to alleviate oxidative stress, ultimately preserving functional muscle tissue. The effect of pGz on ROS was mediated by NO, since pre-treatment with L-NAME completely nullified the effect induced by pGz on intracellular ROS production.

### 4.3. Reduction of Muscle Damage by pGz in db/db Mice

The serum level of CK in db/db mice was significantly higher (834%) compared to WT mice, indicating muscle damage. This finding is consistent with previous studies that reported elevated CK levels in diabetic subjects compared to individuals without diabetes [44,45]. pGz treatment in db/db mice decreased CK levels (52%), demonstrating a reduction in muscle damage. This finding is consistent with previous studies that have demonstrated the beneficial effects of pGz treatment on muscle function and integrity.

### 4.4. Decrease Inflammatory Cytokines by pGz

We found that IL-6 and TNF-α were increased in db/db mice compared to WT. Inflammatory cytokines, such as IL-6 and TNF-α, have emerged as key players in the pathogenesis of T2D [46,47,48,49]. In the context of T2D, IL-6 and TNF-α are known to be upregulated, contributing to insulin resistance, impaired insulin-mediated glucose uptake in peripheral tissues [50], and endothelial dysfunction in T2D [51]. Furthermore, TNF-α is known to down-regulate eNOS expression and activity at the post-transcriptional level, contributing to coronary endothelial dysfunction in T2D mice [51]. pGz significantly reduced plasma IL-6 and TNF-α in db/db mice, with no significant effect on WT mice. The mechanism by which pGz exerts its anti-inflammatory effects, leading to a reduction in IL-6 and TNF-α levels in db/db mice, can be attributed to a reduction in iNOS expression, which is typically upregulated under inflammatory conditions. Furthermore, pGz increases the expression of eNOS and nNOS, both of which are involved in the production of NO with anti-inflammatory properties [52]. By modulating these inflammatory pathways, it may be possible to improve insulin sensitivity and glucose homeostasis in diabetic patients. In this context, pGz represents an alternative approach that has shown promise in modulating inflammatory signaling and improving metabolic health in various disease models [4,52]. However, further research is needed in order to explore the specific effects of pGz on IL-6 and TNF-α signaling in the context of T2D management.

### 4.5. pGz Reduces Muscle Fatigability and Increases Strength in db/db Mice

The current study revealed significant differences in fatigue susceptibility and force generation between the db/db and WT groups. This increased susceptibility to fatigue and reduced strength aligned with observations in patients with TD2, who often exhibit reduced muscle endurance and more significant fatigue during physical activity [53,54,55]. However, when db/db mice were subjected to pGz application, significant improvements in both fatigue resistance and muscle force were observed. The exact mechanisms by which pGz affects skeletal muscle in the context of T2D are not yet completely understood. However, they may be associated with the ability of pGz to (i) decrease the elevated [Ca^2+^]_i_ in skeletal muscle fibers [4], and (ii) reduce oxidative stress and chronic inflammation [52]. Furthermore, impaired blood flow and vascular dysfunction are common in T2D muscles [56], and NO acts as a potent vasodilator, optimizing the perfusion of skeletal muscle and ensuring an adequate supply of oxygen and energy substrates [57].

### 4.6. Effects of pGz on NOS Expression in db/db Muscles

NO is a gaseous signaling molecule that plays a vital role in diverse functions, including anti-inflammatory effects [58]. Since NO is a short-lived molecule, its regulation occurs mainly at the level of NO synthesis from L-arginine. NO synthesis and regulation involve constitutively expressed nNOS and eNOS that are responsible for producing relatively small amounts of NO (nanomolar range), which have a protective effect on cells. In addition, there is an inducible iNOS, which is a robust NO producer, generating large amounts of NO (micromolar range), which can have toxic effects on cells [58]. In specific tissues, such as skeletal muscle, NO synthesis by constitutive NOS enzymes is vital for insulin action [59,60,61]. The results of the current investigation revealed a significant deficit in eNOS and p-eNOS expression in the skeletal muscle of db/db mice compared to WT. eNOS expression deficiency has been closely associated with insulin resistance [62]. Interestingly, using genetically modified mice with ablation of both eNOS and nNOS also led to the development of insulin resistance [62]. The administration of pGz treatment resulted in a notable increase in the expression of eNOS, p-eNOS, and eNOS in both WT and db/db skeletal muscle.

iNOS plays a crucial role in mediating inflammation and has emerged as a significant contributor to insulin resistance, a primary causative factor for T2D [63]. We found that iNOS expression was significantly elevated in the muscle of db/db mice compared to WT, where its expression is almost undetectable, a finding that agrees with a previous report [64]. Furthermore, human studies have also shown that iNOS expression increases in the skeletal muscle of individuals with T2D [65]. Therefore, increased expression of iNOS and subsequent overproduction of NO may cause enhanced S-nitrosylation of the ryanodine receptors (RyR) [66] of the insulin receptor substrate-1 and protein kinase B (AKT) [67,68]. In particular, protein S-nitrosylation has been reported to be elevated in patients with T2D [68]. In the current report, we found that pGz significantly reduced iNOS expression, and this reduction was associated with an improvement in muscle insulin-dependent glucose uptake, a reduction in [Ca^2+^]_i_ and ROS production, decreased plasma concentrations of IL-6 and TNF-α, and improvements in muscle function. Although the exact mechanism by which pGz reduces muscle iNOS remains unknown, it is believed that the physiological levels of NO produced by eNOS and nNOS, both enhanced by pGz, inhibited iNOS transcription through the inactivation of nuclear factor-KB (NF-KB) [69].

However, despite our primary focus being on skeletal muscle, the effect of pGz on T2D may also be mediated through organs beyond the skeletal muscle, such as the liver, which plays a crucial role in regulating glucose metabolism [23]. This hypothesis has support from several observations: first, NO is a critical player in carbohydrate metabolism, and reduced NO bioavailability is associated with T2D [70,71,72,73]. pGz has been shown to augment blood circulation and raise eNO levels across multiple organs, including the liver [74]. Second, the inhibition of iNOS has shown the potential to reverse or improve obesity-induced insulin resistance in both the skeletal muscle and the liver in mouse models [68]. Liver-specific activation of NFĸβ has been demonstrated to induce hepatic insulin resistance in mice [75], and pGz has been demonstrated to reduce iNOS and NFĸβ expression [76,77].

These findings are consistent with earlier research, indicating that inhibition of iNOS through gene manipulation or pharmacological inhibitors can prevent or ameliorate insulin resistance in the skeletal muscle of rodents [67]. The ability of pGz to enhance the expression of eNOS, p-eNOS, and nNOS and to reduce iNOS in the skeletal muscle of db/db mice suggests that it may hold therapeutic potential for the management of patients with T2D.

### 4.7. Study Limitations

Despite the novel findings obtained in this study, it is essential to acknowledge certain limitations. Firstly, our research concentrated on a specific age range of db/db mice, specifically those aged 6 months. It is essential to recognize that T2D is typically low in younger individuals [78]. Secondly, it is important to note that our study did not investigate the effects of pGz treatment beyond 8 days. Therefore, the potential benefits of extending the treatment duration remain unknown. Thirdly, it is essential to point out that this study did not investigate the specific mechanism by which pGz reduces muscle intracellular [Ca^2+^], such as its effect on sarco/endoplasmic reticulum ATPase, the Na^+^/Ca^2+^ exchanger, and the plasma membrane ATPase.

## 5. Conclusions

The present study confirms that db/db muscle shows elevated [Ca^2+^]_i_ and reduced glucose uptake. In addition, it provides evidence of the beneficial effects of passive pulsatile shear stress, as provided by pGz on muscles with insulin resistance. This beneficial effect seems to be mediated by reducing [Ca^2+^]_i_, alleviating muscle oxidative stress, reducing proinflammatory cytokines, increasing the expression of eNOS and nNOS, reducing iNOS and muscle injury, and improving muscle function. These findings underscore the promising therapeutic effects of passive pulsatile shear stress on muscle-related complications associated with T2D. More research is needed in order to explore the underlying mechanisms and translate these findings into clinical applications for patients suffering from T2D.

## Figures and Tables

**Figure 1 biomedicines-11-02596-f001:**
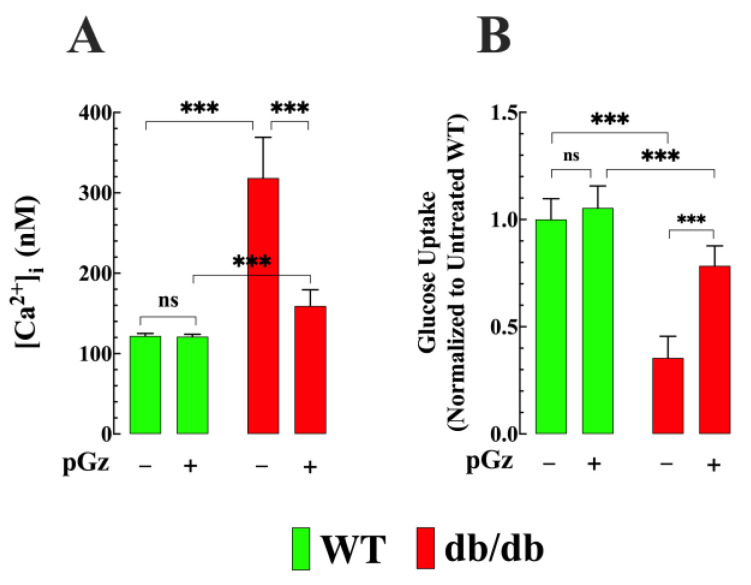
The effect of pGz on [Ca^2+^]_i_ and glucose uptake in WT and db/db muscle cells. (**A**) In db/db muscle cells (*n* = 20 and *n_mic_* = 3), the average [Ca^2+^]_i_ was 318 ± 51 nM, while in WT, it was 122 ± 3 nM (*n* = 16 and *n_mic_* = 3). (**B**) Glucose uptake was reduced in db/db muscles (*n* = 21 and *n_mic_* = 3) compared to WT muscle fibers. Pretreatment of db/db mice with pGz resulted in a significant reduction in muscle [Ca^2+^]_i_ (*n* = 20 and *n_mice_* = 3); shown in (**A**) is the increase in glucose uptake in db/db muscle cells (*n* = 21 and *n_mice_* = 3) due to pGz treatment compared to untreated db/db mice. No significant effect on [Ca^2+^]_i_ or glucose uptake was observed in the WT group (**A**,**B**). Values are expressed as the mean ± SD. Statistical analysis was performed using one-way ANOVA followed by Tukey’s post hoc comparisons; ns denotes *p* > 0.05 and *** *p* ≤ 0.001.

**Figure 2 biomedicines-11-02596-f002:**
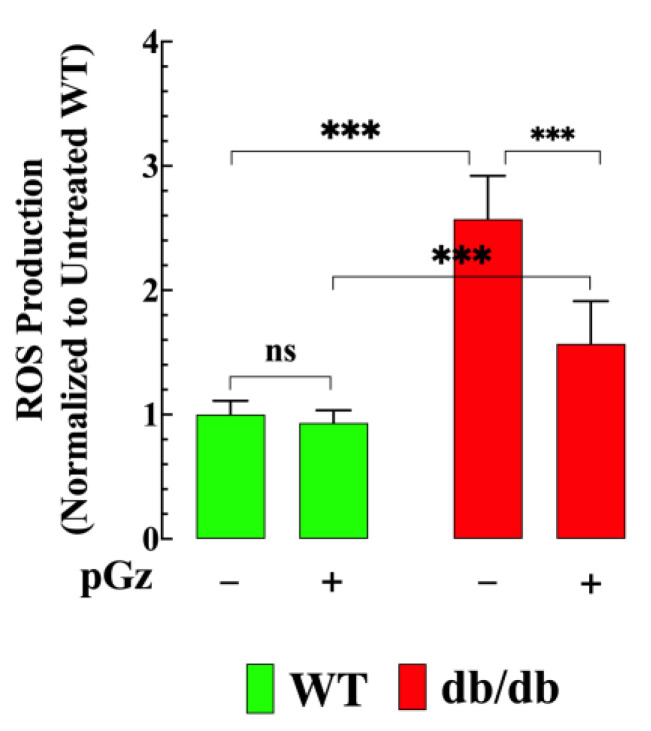
pGz reduces ROS production in db/db muscle cells. ROS production was significantly increased in db/db muscle fibers (*n* = 17 and *n_mice_* = 3) compared to WT muscle fibers (*n* = 15 and *n_mice_* = 3). pGz significantly reduced intracellular ROS in db/db muscle cells (*n* = 18 and *n_mice_* = 3). No effect of pGz was detected in WT muscle fibers (*n* = 16 and *n_mice_* = 3). Values are expressed as mean ± S.D. Statistical analysis was performed as described in Figure 1**,** with ns indicating no statistical significance and *** indicating *p* < 0.001.

**Figure 3 biomedicines-11-02596-f003:**
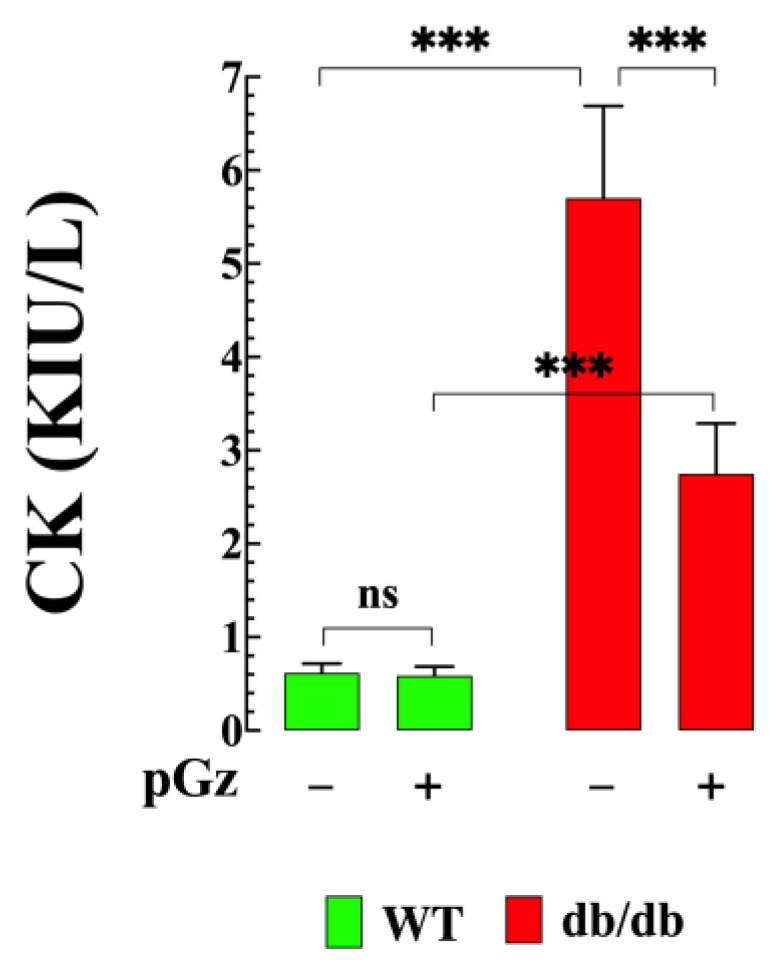
pGz diminishes muscle damage. Creatine kinase measurements were markedly elevated in db/db mice (n = 6 and *n_mice_* = 6) compared to WT mice (*n* = 6 and *n_mice_* = 6). pGz treatment significantly reduced CK levels in db/db mice (*n* = 6 and *n_mice_* = 6), but it had no effect in WT mice (*n* = 6 and *n_mice_* = 6). All values are presented as the mean ± SD. CK activity is expressed as Kilo International Units per liter (KIU/L). Statistical analysis was performed following the methods described in Figure 1, with ns indicating no statistical significance and *** indicating *p* < 0.001.

**Figure 4 biomedicines-11-02596-f004:**
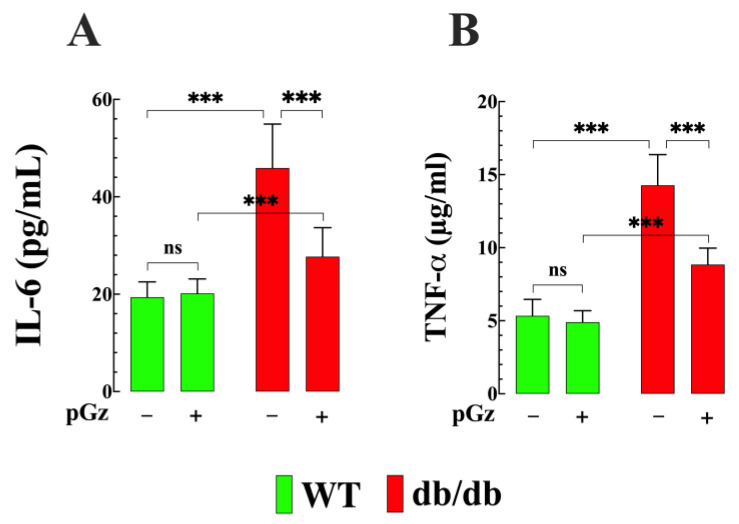
Gz reduces pro-inflammatory cytokines in db/db mice. (**A**) The plasma IL-6 concentration increased in db/db mice (*n* = 12 and *n_mice_* = 6) compared to WT mice (*n* = 10 and *n_mice_* = 5). Treatment with pGz significantly reduced the plasma levels of IL-6 in db/db mice (*n* = 14 and *n_mice_*= 7), while no effect was observed in WT (*n* = 12 and *n_mice_* = 6). (**B**) The plasma TNF-α concentration was elevated in db/db mice (*n* = 12 and *n_mice_* = 6). pGz reduced the TNF-α levels in db/db mice (*n* = 14 and *n_mice_* = 7. No significant effect was observed in WT mice (*n* = 12 and *n_mice_* = 6). All values are presented as the mean ± SD. Statistical analysis was performed as described in Figure 1; ns denotes *p* > 0.05 and *** *p* ≤ 0.001.

**Figure 5 biomedicines-11-02596-f005:**
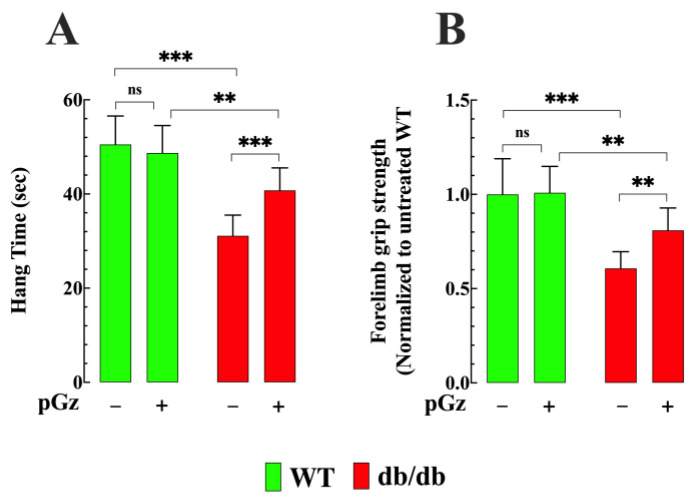
Effects of pGz on muscle function. (**A**) db/db mice (*n* = 30 and *n_mice_* = 10) exhibited reduced hanging times compared to WT mice (*n* = 24 and *n_mice_* = 8). Treatment with pGz increased the hanging time in db/db mice (*n* = 39 and *n_mice_* = 13). (**B**) The grip strength of the forelimb was reduced in db/db mice (*n* = 33 and *n_mice_* = 11) compared to WT mice. pGz treatment improved muscle strength in db/db mice (*n*= 33 and *n_mice_* = 11). No significant effect was observed in WT mice after treatment with pGz. All values are presented as the mean ± SD. Statistical analysis was performed as described in Figure 1; ns denotes *p* > 0.05 and ** *p* < 0.01, *** *p* ≤ 0.001.

**Figure 6 biomedicines-11-02596-f006:**
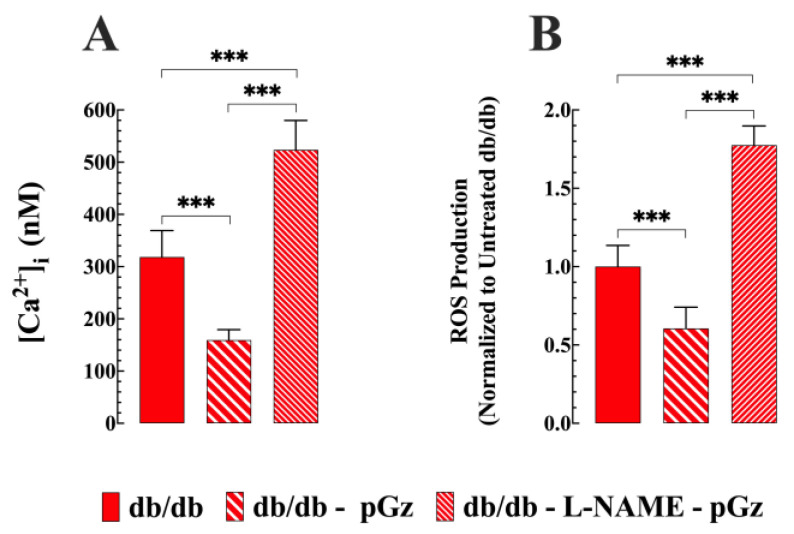
NO mediates the cytoprotective effects of pGz. Pretreatment of db/db mice with L-NAME not only abolished the protective effects of pGz on [Ca^2+^]_i_ levels and ROS production, but also exacerbated the detrimental effects in db/db muscle cells. (**A**) [Ca^2+^]_i_ in db/db muscles was further elevated (*n* = 17 and *n_mice_* = 4) compared to the untreated db/db muscle fibers. (**B**) Similarly, L-NAME treatment caused an intracellular ROS elevation in db/db muscle cells (*n* = 18 and *n**_mice_* = 4) compared to untreated db/db. All values are presented as mean ± SD. Statistical analysis was performed as described in Figure 1; *** *p* ≤ 0.001.

**Figure 7 biomedicines-11-02596-f007:**
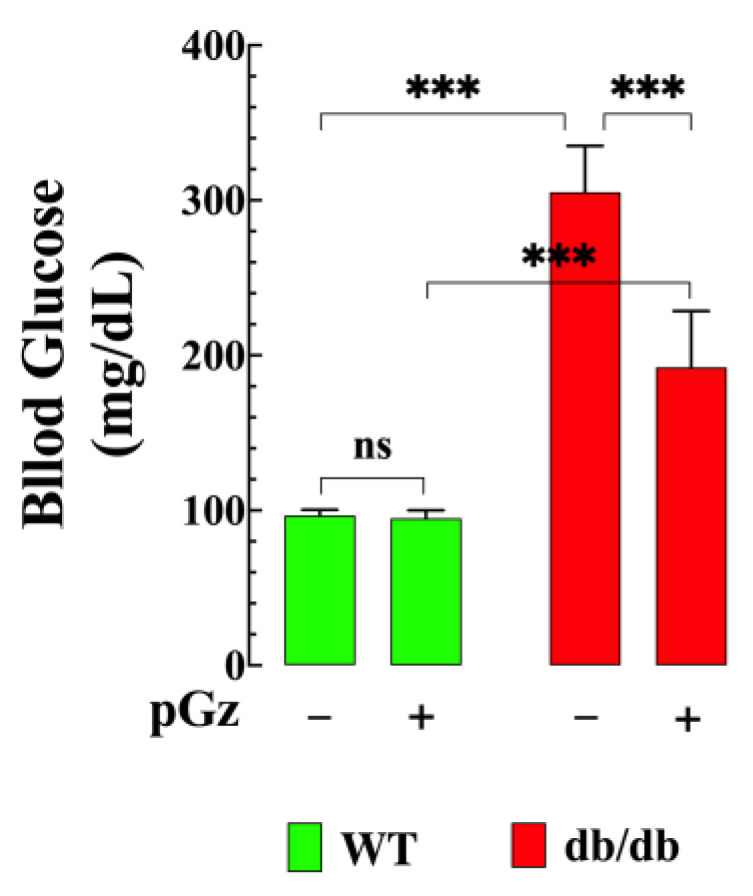
pGz reduced hyperglycemia in db/db mice. Fasting untreated db/db mice (*n* = 18 and *n_mice_* = 9) exhibited blood glucose levels 260% higher than untreated WT mice (*n* = 16 and *n_mice_* = 8). However, pGz treatment led to a significant reduction of 45% in the blood glucose levels of db/db mice (*n* = 18 and *n_mice_* = 9). In contrast, pGz treatment did not significantly affect the blood glucose levels of WT mice (*n* = 16 and *n_mice_* = 8). All values are expressed as means ± S.D. The statistical analysis was carried out as described in Figure 1, with ns indicating no statistical significance and *** indicating *p* < 0.001.

**Figure 8 biomedicines-11-02596-f008:**
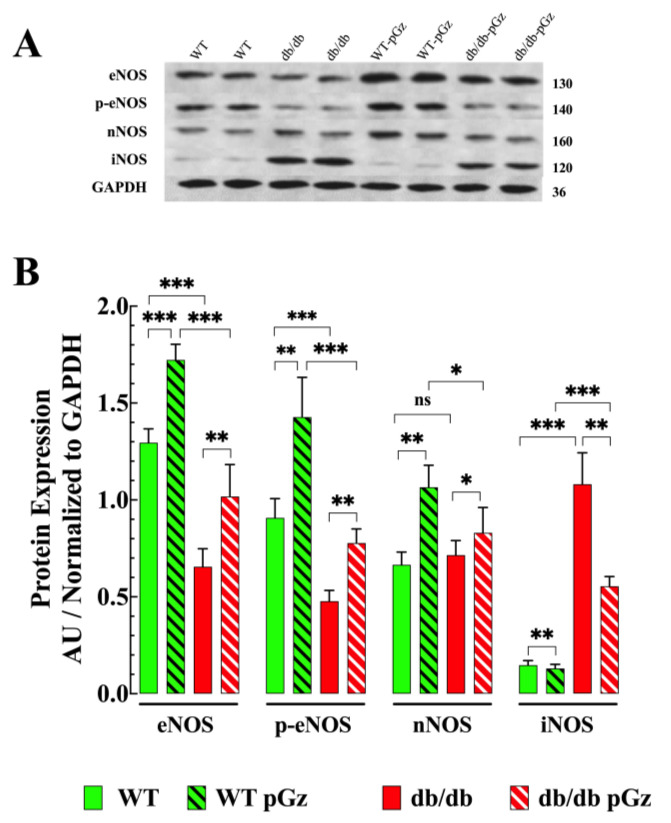
Effects of pGz on protein expression. (**A**) Representative Western blot analysis of the expression of the eNOS, p-eNOS, nNOS, and iNOS proteins in WT and db/db skeletal muscles. The right axis shows the molecular weight of the proteins. (**B**) Densitometric analysis of 4 independent Western blot experiments. Data were normalized to GAPDH and expressed as mean ± S.D. Paired *t*-test; ns denotes *p* > 0.05; ** p* < 0.05, ** *p* < 0.01, *** *p* < 0.001.

## Data Availability

The raw data supporting the conclusions of this manuscript will be made available by the authors, without undue reservation, to any qualified researcher.

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
