# Peer review of "Enhancing Muscle Intracellular Ca2+ Homeostasis and Glucose Uptake: Passive Pulsatile Shear Stress Treatment in Type 2 Diabetes"

_biomedicines, 2023, doi:10.3390/biomedicines11102596_

Round 1

Reviewer 1 Report

The authors aimed to investigate the effect of pGz on db/db a rodent model of T2D. Treatment of db/db mice with pGz resulted in several beneficial effects. It reduced [Ca2+]i overload, enhanced muscle glucose transport, and decreased ROS levels, interleukins, and CK. Furthermore, pGz treatment increased the expression of endothelial nitric oxide synthase (eNOS), phosphorylated eNOS (p-eNOS), and neuronal nitric oxide synthase (nNOS) and reduced inducible nitric oxide synthase (iNOS), and improved muscle strength.

The subject of manuscript is very interesting and in the scope of the journal, bringing an important issue for the new potential therapeutics for type 2 diabetes. Although the authors' consent and observations are interesting there are some points that require further justification.

Major comments:

Why did you measure any anti-inflammatory cytokine?

The AMPK mediated glucose uptake should also be evaluated. Because it seems that pGz has a similar molecular effect of exercise, so this pathway can be involved in the beneficial effects.

How was the insulin resistance/pathway in this animals?

The effect is only observed in muscle cells, or in other organs? This should be discussed.

Minor comments:

A reference should be cited in the sentence in line 43-46 in the introduction.

Please describe the day time for the pGz protocol.

Author Response

Reviewer #1

Major comments:

1.- Why did you measure any anti-inflammatory cytokine?

Response: In prior studies, we demonstrated that pGz successfully reduced the elevation of inflammatory cytokines, including TNF-α, MCP-1, and IL-6, in skeletal muscle subjected to eccentric exercise (PMID: 27031739). Moreover, pGz also mitigated the increase in pro-inflammatory cytokines during an LPS-induced cytokine storm in cardiomyocytes (PMC7970274

2.- The AMPK mediated glucose uptake should also be evaluated. Because it seems that pGz has a similar molecular effect of exercise, so this pathway can be involved in the beneficial effects.

Response: AMPK activity, typically stimulated by factors like muscle contraction, enhances muscle glucose transport independently of insulin (PMID: 16822958). Muscle contraction raises [Ca2+]i levels, initiating glucose transport (PMID: 12556346 and PMID: 16822958). In our study, pGz was applied to quiescent muscle fibers, devoid of muscle contractions and [Ca2+]i elevation; instead, pGz reduced [Ca2+]i . Under these specific conditions, it seems unlikely that AMPK plays a role in pGz-induced glucose transport.

Our findings suggest two possible mechanisms:

  1. i) Nitric oxide (NO) can stimulate glucose uptake independently of insulin through NOS activation (PMID: 7527495). Our study reveals that pGz increases the expression of both eNOS and nNOS in muscle cells, implying that pGz may boost glucose uptake by elevating NO production.
  2. ii) Alternatively, there may be crosstalk between [Ca2+]i and glucose uptake in skeletal muscle (PMID: 35547584). Elevated [Ca2+]i i, as seen in db/db muscle cells, impairs insulin-stimulated glucose uptake, disrupting glucose homeostasis. Conversely, reducing chronic [Ca2+]i elevation, as achieved with pGz, may facilitate glucose uptake and mitigate hyperglycemia."

3.- How was the insulin resistance/pathway in this animals?

Response: In a previous study (PMID 35547584), we conducted an analysis of plasma insulin responses following glucose stimulation in db/db mice. These mice exhibited markedly elevated fasting insulin levels, approximately 4.7 times higher than their WT counterparts. Furthermore, after the administration of a glucose load, the peak insulin levels at 30 minutes post-stimulation were observed to be approximately twice as high in db/db mice compared to WT mice. Additionally, the rate of decline in blood insulin levels at 60, 90, and 120 minutes following stimulation was significantly slower in db/db mice when compared to WT mice. Moreover, the area under the curve for insulin levels over the 0–120 minute period was notably greater in db/db mice as compared to their WT counterparts

4.- The effect is only observed in muscle cells, or in other organs? This should be discussed.

Response: Our research has encompassed a comprehensive examination of pGz's impact on various tissues, including skeletal, smooth, and cardiac cells, as well as cortical and hippocampal neurons. However, our specific investigation into the influence of pGz on glucose transport has been confined to skeletal muscle. This focus stems from skeletal muscle's critical role as the primary regulator of glucose homeostasis, responsible for approximately 80% of postprandial glucose uptake from the bloodstream. Furthermore, given that muscle cells are central to glucose uptake and that disruptions in this function, such as insulin resistance, are the hallmark characteristics of type 2 diabetes and play a pivotal role in the disease's pathogenesis (PMID: 32940941).

Minor comments:

1.- A reference should be cited in the sentence in lines 43-46 in the introduction.

Response: The requested reference was added (Uryash et al., 2022) (line 47).

2.- Please describe the daytime for the pGz protocol.

Response: The time of the day (morning 9-11 am) has been added (line 81)

Reviewer 2 Report

This manuscript by Arkady Uryash et al., presents a “ Enhancing Muscle Intracellular Ca2+ Homeostasis and Glucose 2 Uptake: Passive Pulsatile Shear Stress Treatment in Type 2 Diabetes. 

The authors emphasized the pGz effect on the diabetes murine model. 

Here is where I have minor concerns. During static analysis why not compare pGZ effect between WT and db/db mice. Fig1, Fig2, Fig3, Fig4, Fig5, and Fig 7.

The authors should check it.  

Inside the Abstract, for calcium 2+ make a superscript.

Lane 28: add + one Ca2

Line 35: “constitute the hallmark” need Italic?

Line 62: muscle cells

Line 123: font make an Italic.

Line 160: db/db

Line 195: need subscript for O2 and CO2

Line 210:  “in vivo” font to Italic

Line 219:  double check of “2.2 time” 318/159= 2 fold only not a 2.2 time. How about say reduced 50%

Line 232: Italic

Line 236, 237, and 238: KIU/L

Figur3  : Y-axis need (KIU/L)

Line 256: remove “ ( “

Figure 7: please double check for 3.1 times and 1.6 times. My calculation is 3.6 fold and 1.8 fold different 

Line 299 remove words “and 1.6 (p<0.01 compared to untreated db/db) was observed”

Line 300: ” treatme” change to treat

Figure 8: A: Give the MW on beside a each image,  B: what is the Y-Axis?

Line 330: “[Ca2+]I” change to [Ca2+]i

Author Response

Reviewer #2

1.- During static analysis why not compare pGZ effect between WT and db/db mice. Fig1, Fig2, Fig3, Fig4, Fig5, and Fig 7.The authors should check it.  

Response: Fig1, Fig2, Fig3, Fig4, Fig5, and Fig 7 are now presented as requested by the Editor and Reviewer.

2.- Inside the Abstract, for calcium 2+ make a superscript.

Response: The superscript has been corrected (line 21).

3.- Lane 28: add + one Ca2

Response: The + sign was added (line 28).

4.- Line 35: “constitute the hallmark” need Italic.

Response: Correction has been made (line 36).

5.- Line 62: muscle cells

Response: The correction has been made (line 63).

6.- Line 123: font make an Italic

Response: The correction has been made (line 128).

Line 160: db/db

Response: The correction has been made (line 165).

7.-Line 195: need subscript for O2 and CO2

Response: The correction has been made (line 200).

8.- Line 210:  “in vivo” font to Italic

Response: The correction has been made (line 219).

9.- Line 219:  double check of “2.2 time” 318/159= 2 fold only not a 2.2 time. How about say reduced 50%

Response: We thank the reviewer for the observation, the correction has been made and now we use % instead of X-time in the manuscript.

10.- Line 232: Italic

Response: The correction has been made

.

11.- Line 236, 237, and 238: KIU/L

Response: The correction has been made (line 260)

12.- Figur3  : Y-axis need (KIU/L).

Response: The correction has been made in the Figure 3 legend and the manuscript text (line 260)

13.- Line 256: remove

Response: The correction has been made

14.- Figure 7: please double check for 3.1 times and 1.6 times. My calculation is 3.6 fold and 1.8 fold different 

Response: We thank the reviewer for the observation and the correction has been made

15.- Line 299 remove words “and 1.6 (p<0.01 compared to untreated db/db) was observed”

Response: The correction has been made

16.- Line 300: ” treatme” change to treat

Response: The correction has been made (line 314)

17.- Figure 8: A: Give the MW on beside a each image,  B: what is the Y-Axis?

Response: The MW has been added to the bands and the label of Y-Axis has been added to Figure 8.

18.- Line 330: “[Ca2+]I” change to [Ca2+]i

Response: The correction has been made

Round 2

Reviewer 1 Report

Thanks for the anwers! However when I asked about the insulin resistance in the animals, I was asking in the animals submited to the pGz. Can you please add some insights in the discussion?

Another thing is that although the focus of the study was skeletal muscle, you should at least discuss in one paragraph the potential effect of pGz in other tissues, for instance liver, which as a major role in glucose homeostasis. 
